# Analysis of the Motivation of Students of the Last Cycle of Primary School in the Subject of Physical Education

**DOI:** 10.3390/ijerph19031332

**Published:** 2022-01-25

**Authors:** Jorge Rojo-Ramos, María José González-Becerra, Santiago Gómez-Paniagua, Eugenio Merellano-Navarro, José Carmelo Adsuar

**Affiliations:** 1Motricity and Education (HEME) Research Group, Department of Health, Economy, University of Extremadura, Avda. de la Universidad s/n, 10003 Cáceres, Spain; jadssal@unex.es; 2BioẼrgon Research Group, University of Extremadura, 10003 Cáceres, Spain; mgonzalexd@alumnos.unex.es (M.J.G.-B.); sgomezpa@alumnos.unex.es (S.G.-P.); 3Grupo de Investigación EFISAL, Universidad Autónoma de Chile, Talca 3460000, Chile

**Keywords:** motivation, physical education, school

## Abstract

Motivation is the impulse that leads people to perform certain actions and persist in them to achieve certain objectives. Motivation is important in all areas of life and has a very important role in academics, where it can be considered essential for learning motor skills and performance. In this study, we intended to evaluate motivation in the Physical Education classroom among students in the last grade of elementary school. For this purpose, a total of 545 students aged between 9 and 13 years were selected to take the CMEF-EP questionnaire through a tablet and by means of the Google Forms application. The different relationships between items and dimensions were analyzed using the Mann–Whitney U test. A Spearman’s test was used to explore the relationship between dimensions and age. The results showed that the predominant motivation in the classroom is intrinsic motivation and that demotivation is practically nonexistent. In addition, it was found that there are differences between the genders in some of the categories and that there are no differences in the location of the study center. Therefore, it can be said that it is essential to generate a motivational climate based on the subject and his or her interests in order to favor intrinsic motivation.

## 1. Introduction

The many benefits of physical activity (PA) are well known, whether at a physical, psychological or social level. For this reason, it has become an aspiration for today’s societies to promote physical education that is capable of promoting healthy lifestyles. With this purpose in mind, the World Health Organization (WHO) makes recommendations regarding the amount of daily physical activity that should be carried out by children, being at least 60 min a day of aerobic activity of moderate or vigorous intensity. Children and adolescents spend most of their time in educational centers, which become their social environment and generate an important part of their influences, such as friendships or opportunities for PA and the probability of continuing to do it outside of the educational environment [1]. Promoting messages in schools that doing some physical activity is better than remaining inactive, or that a higher level of sedentary lifestyle is associated with poor health outcomes such as greater adiposity, poorer cardiometabolic health, physical fitness and prosocial behavior, and shorter sleep duration, is a practice that should be taken into account to start from the base to generate adherence to the practice of physical activity recommended by the WHO.

At school age, Physical Education is a fundamental discipline in the education and integral formation of children, offering them the possibility of developing motor, cognitive and affective skills that are essential for their lives. Through these classes, children learn, execute and create new forms of movement through different forms of play, recreation and sports, thus encouraging creativity and spontaneity and allowing them to know, respect and value themselves and those around them. Giakoni Ramírez et al. point out that performing at least four hours of PE per week has a positive effect on physical fitness and body composition factors, increasing PA levels from moderate to vigorous [2]. Rodríguez Torres et al. recognize, in their study, the influence of physical activity on mental health, especially the reduction of anxiety and stress [1]. Contreras-Jordan et al. emphasize the advantages of Physical Education (PE) at the cognitive level, relating PA to academic performance, and on sedentarism [3]. Specifically, the most visible adaptations that occur after sports practice are recognized in self-esteem, emotions and motor skills.

Physical Education classes have played a very important role over the years within the educational context, considering PE and the motivation generated as a direct influence on fostering healthy lifestyle habits and also generating adherence to the practice of Physical Activity in the out-of-school context [4]. However, motivation is not only important for PE, but can also be related to school failure, considering motivation as a hypothetical construct that explains the initiation, direction and perseverance of a behavior towards a certain academic goal focused on learning, performance, self, social valuation or work avoidance [5]. In the academic context, achieving goals is not only a matter of having the necessary skills and knowledge, but also a matter of willingness and motivation [6]. Following Järvenoja, motivation is considered to be the active process by which a person attends to and stays with a task in order to achieve its purposes [7]. At different levels of sport and in PA, motivation plays a fundamental role in learning, the acquisition of motor skills and performance, contributing to the valuing of self-effort and perseverance as elements that can influence the achievement of personal satisfaction [4].

It can be said that there are two sources of motivation: extrinsic motivation, which involves working with the purpose of obtaining rewards from external sources [8] such as getting good grades, earning money or pleasing someone; and intrinsic motivation, which, on the contrary, involves performing activities for the mere sensation of pleasure, enjoyment, interest or satisfaction. In addition to these sources of motivation, it is worth knowing that amotivation or demotivation represents the absence of any form of motivation [8]. In their study, Navarro-Patón et al. establish that intrinsic motivation positively predicts the degree of enjoyment of physical education. In this sense, if the student achieves a motivation for the practice of PA based on intrinsic arguments, it is very likely that in the future he/she will achieve a high adherence to the practice of physical sports activity in the future [9]. Motivation does not only have consequences on sports practice, but can also be reflected at the cognitive level. According to Vallerand, it can have positive consequences, such as effort or perseverance, related to intrinsic motivation, or more negative ones such as boredom, related to extrinsic motivation [10]. This should be an opportunity for the PE teacher to take timely and necessary measures in the teaching–learning process, which can be achieved through the use of different methodologies, positive feedback or variety in the tasks [8]. The Flipped Classroom (FC), or inverted classroom, is presented as an educational methodology for ICT work that promotes the active participation of the student in the generation of knowledge [11]. According to Iborra Urios et al., the use of this strategy promotes self-learning and autonomous work and achieves a deeper understanding of the content [12]. Francés et al. also used the inverted classroom in their study through the Moodle platform to assess the improvement of motivation and the students’ perception of the subject [13]. For its part, gamification consists of the use of game mechanics in non-game environments, resulting in an opportunity to work on motivation or effort [14].

The Achievement Goals Theory (AGT) provides a cognitive–social approach to study and understand motivation. According to this theory, a person is ego-oriented when they use references to evaluate their success, and task-oriented when they take personal references to evaluate their success or failure [15]. The results of previous studies affirm that the motivational climate in PE has a relationship with AGT. This is associated with a more task-oriented motivational climate and a negative relationship with an ego-oriented motivational climate [4]. Other studies, such as that of Gråstén and Watt, determine that boys and girls experience positive effects on cognition, enjoyment and practice of out-of-school PA when there is a task-oriented climate, while an ego-oriented climate generates negative effects [16].

In the present study, we aimed to evaluate the levels of motivation in the physical education classroom among elementary school students, especially the association between the different items, the differences between gender and school location, and the relationship between age and the different types of motivation. For this purpose, the Motivation in Physical Education Questionnaire (CMEF) was used as a data collection instrument. This scale is composed of the initial sentence “I participate in Physical Education classes…” and followed by 20 items that analyze intrinsic motivation, identified regulation, introjected regulation, extrinsic regulation and demotivation. As this questionnaire was designed for high school students, it should be noted that there may be differences in the results, but it is still valid, as other articles have already shown [17]. The last cycle of primary school was selected because children in this year group are at a key age to work on motivation towards Physical Education, since it is framed just after Infant Education, where the main motivation of children is to play, and moves into Secondary Education, a stage where they begin to develop as people and start to perform activities that move them away from healthy life habits and, therefore, from Physical Education. In their study, Leo et al. demonstrate the psychometric properties of the CMEF with elementary school students, which allows this instrument to be used effectively with students of these ages [18].

## 2. Materials and Methods

### 2.1. Participants

The participants were selected using a non-probabilistic sampling method based on convenience sampling [19]. The sample consisted of a total of 545 Physical Education students in the last cycle of the primary stage, of whom 50.8% (277) were male and 49.2% (268) were female, aged between 9 and 13 years (M = 11.04; SD = 0.80), all of them belonging to public or private-subsidized schools in the region of Extremadura (Spain).

Table 1 shows the distribution of the participants according to gender, type of school, grade, province, school environment and age.

### 2.2. Instruments and Measures

Sociodemographic data were obtained from a form designed using Google Forms with six sociodemographic questions (gender, type of center, grade, province, center environment and age).

The Motivation in Physical Education in Primary Education (CMEF-EP) questionnaire [18] was used. This instrument uses a Likert scale, where values range from 1 to 5, with 1 being “Strongly disagree” and 5 being “Strongly agree”. The authors reported acceptable internal consistency indices greater than 0.70 for all factors.

The CMEF-EP consists of 18 items, grouped into 5 factors. Factor 1 “Intrinsic motivation” (composed of 4 items), factor 2 “Identified regulation” (composed of 4 items); factor 3 “Introjected regulation” (composed of 2 items); factor 4 “External motivation” (composed of 4 items) and factor 5 “Demotivation” (composed of 4 items).

### 2.3. Procedures

A digital questionnaire was developed using the Google Forms application incorporating the sociodemographic questions and the CMEF-EP. This format was chosen as it allowed us to save costs, paper, time and obtain higher return rates [20]. The data collection was carried out during the months of March, April and May 2021.

To access the sample, we first used the database of educational centers of the Department of Education and Employment of the Regional Government of Extremadura (https://ciudadano.gobex.es/ciudadanoportlet/printpdf/pdf?typepdf=3443&idDirectorio=775, accessed on 1 March 2021) and sent an e-mail to the directors of all of the centers informing them of the purpose of the study, accompanied by informed consent and an invitation to collaborate in the study. Those schools that agreed to participate in the study provided us with a date on which we could go to the classrooms in person so that the students could complete the questionnaire on tablets that we provided them with to facilitate the procedure and avoid technical or resource problems. All data were collected anonymously and the average response time was 5 min.

### 2.4. Statistical Analysis

The analysis of the data collected was performed with the Statistical Package for Social Sciences (SPSS) version 23.0 for Mac. To analyze whether the variables complied with the assumption of normality, the Kolmogorov–Smirnov test was used, which indicated that this assumption was not met, so the decision was made to use nonparametric tests.

The Mann–Whitney U test was used to analyze the relationships between the items of the different dimensions according to gender and center location (Table 2). The Mann–Whitney U test was also used to analyze the relationships between each of the dimensions as a function of gender and center location (Table 3).

A Spearman’s Rho test was used to explore the relationship between each of the dimensions and the age variable (Table 4).

Reliability was calculated for each of the dimensions using Cronbach’s alpha. Following Nunnally and Bernstein [21], reliability values between 0.60 and 0.70 can be considered acceptable, while values between 0.70 and 0.90 can be considered satisfactory.

## 3. Results

Table 2 shows the associations between the different CMEF-EP items according to gender and center location. The Mann–Whitney U test was used. The scores of the five factors (“Intrinsic”, “Identified”, “Introjected”, “External” and “Demotivation”) were obtained from the median value (Me) of each of the items that made up each dimension. The data are presented as median and interquartile range (IQR).

Table 3 presents the scores obtained in each of the CMEF-EP dimensions according to gender and center location.

In factors 1 (intrinsic motivation) and 3 (external motivation), males obtained higher scores than females, and statistically significant differences were found (*p* < 0.01). In factor 2 (identified regulation), males obtained slightly higher values than females, and significant differences were found according to gender (*p* < 0.01).

No significant differences were found in any of the questionnaire factors as a function of center location.

Table 4 shows the correlations between the factors and age using Spearman’s test.

Finally, the reliability results for each of the dimensions of the CMEF questionnaire were a1 = 0.74; a2 = 0.81; a3 = 0.81; a4 = 0.85; a5 = 0.75, all being satisfactory values above 0.70, according to Nunnally and Bernstein (1994).

## 4. Discussion

This study was conducted to gain a better understanding of motivation towards Physical Education in primary school classrooms, explaining the differences according to gender and school location. The CMEF questionnaire was used to obtain these data. Being motivated to practice sports in schools not only allows better grades to be obtained, but also involves acquiring a series of habits, including adherence to sports practice, which will allow students to lead a healthy life.

For items 1, 6, 11 and 15 in Table 2, which belong to questions related to intrinsic motivation, very high total values belonging to the median were obtained, which informs us that the general intrinsic motivation of the group is high. Studies such as those by García et al., Chen et al. and Navarro Patón et al. [8,22,23] argue that if students are highly motivated between the ages of 10 and 12 years, they are more participative in the activities proposed in Physical Education. In addition, this motivation is produced by their own interest; that is, it would be part of an intrinsic motivation. Thus, the thought that motivation influences the degree of involvement in an activity, which may seem obvious, is also confirmed by Gutiérrez, who relates positive attitudes towards Physical Education, such as valuing the content or considering it useful for development, with feeling intrinsically motivated to engage in physical activity [24]. Other authors who have carried out research on the role of intrinsic motivation in PE classrooms state that it increases when using teaching approaches that focus on the student and his or her personal interests, and that it plays a very important role in academic performance [25,26].

From items 2, 7, 12 and 16 in Table 2, belonging to identified motivation, it can be seen that the overall values of the group are high, which means that most of the students to whom the questionnaire was given identified with the motives imposed on them, and by internalizing them, they created a motivation towards the subject. These results are similar to those of López et al., which put the identified motivation in second place, after intrinsic motivation, as participation in EF to achieve certain personal goals that are important to schoolchildren [27]. In the case of items 3 and 8, which refer to introjected or assumed motivation, it can be seen that the total values are lower than for the previous items, being around 4 out of 5, but with an IQR that shows us that the values are not grouped around 4. This means that the students have not totally assumed this motivation, which is why the results in this case oscillate more and are not grouped around 5, which would indicate being in total agreement. For the questions related to extrinsic motivation, 4, 9, 13 and 17, the total median values are very similar, being close to 4–5 with a small oscillation if we look at the IQR, which in all cases is 2. This is due to a strong need to demonstrate to others that they need to remain motivated to perform the task. Finally, for questions 5, 10, 14 and 18 that refer to demotivation, in almost all cases, the central value is 1; this indicates that the students do not agree with these statements, therefore demonstrating that there is motivation towards the practice of PE. That there is so little demotivation in Physical Education classes is very important, since high levels of demotivation are associated with high levels of frustration [28]. According to Viñado and Mor, students with greater autonomous motivation towards PE and who are less unmotivated are the ones who will be more physically active [29].

Research such as that of López et al. and Ryan and Deci coincide with these results, stating that students in the 3rd grade of Primary Education have a high degree of intrinsic motivation towards PE, with a lower percentage of extrinsic motivation and a low level of demotivation, which leads us to believe that they understand the benefits that PE brings them [27,30]. The *p*-values in the Location column in Table 2 demonstrate that there are no significant differences according to school location. Again, Table 3 shows the results for each dimension of the questionnaire, i.e., for the grouped items. It is observed that the order of the scores from highest to lowest is intrinsic motivation, identified motivation, extrinsic motivation, introjected motivation and demotivation. In their research, Lopez et al. agree with the results for intrinsic and identified motivation, placing them in first and second place, respectively, but differ from the results of the present study in which extrinsic motivation is the third most important, yet introjected motivation is placed third in Lopez et al.’s [27].

Regarding possible differences by gender, it is observed in Table 3 that there are significant differences in the intrinsic, identified and external dimensions with *p*-values < 0.01. In the case of intrinsic motivation, males present higher values than females, a fact already advanced by Mecías et al. when they proposed that 5th and 6th grade male students showed more motivation than girls towards physical activity [31]. Higher scores were also obtained from males in identified motivation, especially in external motivation, where the males tend to be more concerned with being superior in order to meet with people’s approval. Again, in Table 3, no significant differences are found among the factors of the questionnaire related to location.

Finally, Table 4 shows the correlations between the different dimensions of motivation according to the variable age groups. As can be seen, the correlation is significant in the external dimension (*p*-value < 0.01), that is, the age of the subject can be related to extrinsic motivation so that the older or younger the subject, the greater or lesser the motivation caused by external factors such as getting better grades, approval or money.

This study has the limitation of only including a very specific age range of the Primary Education stage and it would be interesting to expand this in future research. It would also be possible for future study to examine the previous and subsequent stages within the region to find out at what point the motivation of the students changes.

## 5. Conclusions

All of the items, except those related to demotivation and introjected motivation, showed significant differences between genders, with the differences being in favor of males. Most of the Physical Education students between 9 and 13 years of age who were surveyed and participated in the Physical Education class were predominantly motivated by self-interest, that is, by intrinsic motivation. Identified, introjected and extrinsic motivation do not play fundamental roles, but were shown to be represented in some students. Demotivation was practically nonexistent, perhaps because the students changed it for another type of motivation different from extrinsic motivation.

Gender differences were observed in the intrinsic, identified and extrinsic dimensions, and in all of them males were favored over females. No relationship was found between age and the different forms of motivation, except for demotivation, which does vary with age.

A strength of this study was the large sample size, being representative of the community we wanted to characterize. In addition, this allowed us to describe practically an entire cycle within Primary Education, the third cycle, which makes this research a novel study.

## Figures and Tables

**Table 1 ijerph-19-01332-t001:** Frequency distribution of the sample (*n* = 545).

Variable	Categories	N/M	%/SD
Gender	Male	277	50.8
	Female	268	49.2
Type of Center	Public	520	95.4
	Private-Subsidized	25	4.6
Course	Fifth grade of primary school	282	51.7
	Sixth grade of primary school	263	48.3
Province	Cáceres	468	85.9
	Badajoz	77	14.1
Center Environment	Rural	303	55.6
	Urban	242	44.4
Age		11.04	0.80

M: mean; SD: standard deviation.

**Table 2 ijerph-19-01332-t002:** Descriptive analysis and differences by gender and center location of the questionnaire items.

Item		Gender			Location		
	Total	Male	Female		Rural	Urban	
I participate in Physical Education classes…	M_e_ (IQR)	M_e_ (IQR)	M_e_ (IQR)	*p*	M_e_ (IQR)	M_e_ (IQR)	*p*
1. Because Physical Education is fun	5 (0)	5 (0)	5 (1)	<0.01 **	5 (1)	5 (0)	0.08
2. Because I can learn skills that I could use in other areas of my life.	5 (1)	5 (1)	5 (1)	<0.01 **	5 (1)	5 (1)	0.51
3. Because I feel bad if I don’t participate in the activities.	4 (3)	4 (3)	4 (3)	0.51	4 (3)	4 (3)	0.43
4. Because it is well regarded by the teacher and peers.	4 (2)	4 (2)	4 (2)	<0.01 **	4 (2)	4 (2)	0.45
5. But I don’t understand why we should have PE.	1 (2)	1 (2)	1 (1)	0.24	1 (2)	1 (2)	0.29
6. Because I find this subject enjoyable and interesting	5 (1)	5 (0)	5 (1)	<0.01 **	5 (1)	5 (1)	0.22
7. Because I value the benefits that this subject can have on my personal development.	5 (1)	5 (1)	5 (1)	<0.01 **	5 (1)	5 (1)	0.76
8. Because I feel bad about myself if I miss class.	4 (3)	4 (3)	4 (2)	0.12	4 (3)	4 (3)	0.64
9. Because I want the teacher to think that I am a good student.	5 (2)	4 (3)	4 (2)	<0.01 **	5 (2)	5 (1)	0.28
10. But I really feel like I’m wasting my time with this subject.	1 (0)	5 (1)	1 (0)	0.37	1 (0)	1 (0)	0.81
11. Because I have fun doing activities	5 (0)	1 (0)	5 (1)	<0.01 **	5 (0)	5 (0)	0.82
12. Because, for me, it is one of the best ways to get useful skills for my future.	5 (1)	5 (0)	4 (2)	<0.01 **	5 (1)	5 (1)	0.42
13. Because I want my peers to value what I do.	4 (2)	5 (1)	4 (2)	<0.01 **	4 (2)	4 (2)	0.53
14. I don’t know; I have the impression that it is useless to continue attending classes.	1 (1)	4 (2)	1 (1)	0.39	1 (1)	1 (1)	0.44
15. For the satisfaction I get from practicing	4 (1)	1 (1)	4 (2)	<0.01 **	5 (1)	4 (1.25)	0.16
16. Because this subject provides me with knowledge and skills that I consider important.	5 (1)	5 (1)	5 (1)	0.01 *	5 (1)	5 (1)	0.02 *
17. To demonstrate my interest in the subject to the teacher and classmates	4 (2)	5 (1)	4 (2)	<0.01 **	4 (2)	4 (1)	0.10
18. I don’t know clearly; because I don’t like it at all.	1 (1)	1 (1)	1 (1)	0.95	1 (1)	1 (1)	0.35

Note: Me = median value; IQR = interquartile range. Note: The correlation is significant in ** *p* < 0.01; * *p* < 0.05. Each dimension score is based on a Likert scale (1–5).

**Table 3 ijerph-19-01332-t003:** Descriptive analysis of each dimension of the questionnaire.

		Gender			Location		
Dimensions	Me (IQR)	Male	Female	*p*	Rural	Urban	*p*
Intrinsic	4.75 (0.25)	4.75 (0.5)	4.5 (1)	<0.01 **	4.75 (0.75)	4.75 (0.75)	0.51
Identified	4.5 (0.5)	4.5 (1)	4.5 (1)	<0.01 **	4.5 (1)	4.5 (1)	0.56
Introduced	3.5 (2.5)	3.5 (2.5)	3.5 (2.5)	0.25	3.5 (2.5)	3.5 (2.5)	0.88
External	4 (1.5)	4.25 (1.13)	3.75 (1.5)	<0.01 **	4 (1.5)	4 (1.5)	0.21
Demotivation	1.25 (1)	1.25 (1)	1.25 (1)	0.34	1.25 (1)	1.25 (1)	0.36

Note: Me = median value; IQR = interquartile range. The correlation is significant in ** *p* < 0.01. Each dimension score is based on a Likert scale (1–5).

**Table 4 ijerph-19-01332-t004:** Correlations between the dimensions and the age group variable.

Dimensions	Age ρ (*p*)
Intrinsic	−0.05 (0.17)
Identified	−0.03 (0.45)
Introduced	−0.05 (0.20)
External	<0.01 (0.88)
Demotivation	0.02 (0.59)

Note: Each score obtained in the dimensions is based on a Likert scale (1–5).

## Data Availability

The datasets used during the current study are available from the corresponding author on reasonable request.

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
