# Peer review of "Analysis of the Motivation of Students of the Last Cycle of Primary School in the Subject of Physical Education"

_ijerph, 2022, doi:10.3390/ijerph19031332_

Round 1

Reviewer 1 Report

Comment 1:

Line 34: The abbreviation (OMS) should be stated in the full name for the first time in the main body of the manuscript.

Comment 2:

The Introduction section should be more concise. Some sentences can be integrated in one paragraph. Otherwise, it would be too long and too hard for readers to follow and understand.

Comment 3:

For the Table 1, it would be better if the author revise “N” into “N/M”, and “%” into “%/SD”. Then, the second line for the “Variable” can be removed. In addition, please add notes for “M” and “SD”.

Comment 4:

Line 135: it is recommended to express in a whole sentence. So, please polish the phrase “Sociodemographic data” into a sentence.

Comment 5:

The formats for Table 2 and Table 3 should be checked again.

Comment 6:

Discuss and compare major findings of this study with previous research, rather than describing the results from tables. Moreover, combine certain sentences, for example, “The p-values in the Location column in Table 2 report that there are no significant differences by school location”, and “Again, Table 3 shows no significant differences in the factors of the questionnaire that are related to location.” into paragraphs would be better.

Author Response

Comment 1:

Line 34: The abbreviation (OMS) should be stated in the full name for the first time in the main body of the manuscript.

Response 1: Thank you, the abbreviation (OMS) has been changed to the full name in English and the abbreviation has been added as this is the first time it appears in the manuscript.

 Comment 2:

The Introduction section should be more concise. Some sentences can be integrated in one paragraph. Otherwise, it would be too long and too hard for readers to follow and understand.

Response 2: Thank you very much for your comment, the introduction section has been modified by creating paragraphs that give more sense to the story and make it easier to follow the reading.

 Comment 3:

For the Table 1, it would be better if the author revise “N” into “N/M”, and “%” into “%/SD”. Then, the second line for the “Variable” can be removed. In addition, please add notes for “M” and “SD”.

Response 3: Thank you very much for your contribution. The first line for "Variable" has been modified by changing "N" to "N/M" and "%" to "%/SD". The second line for "Variable" has been deleted and a note explaining the meaning of "M" and "SD" has been added.

 Comment 4:

Line 135: it is recommended to express in a whole sentence. So, please polish the phrase “Sociodemographic data” into a sentence.

Response 4: The format of the paragraph has been changed, introducing "Sociodemographic data" within the sentence.

 Comment 5:

The formats for Table 2 and Table 3 should be checked again.

Response 5: Thank you very much for your contribution. The formats in Table 2 and Table 3 have been modified to the required format.

 Comment 6:

Discuss and compare major findings of this study with previous research, rather than describing the results from tables. Moreover, combine certain sentences, for example, “The p-values in the Location column in Table 2 report that there are no significant differences by school location”, and “Again, Table 3 shows no significant differences in the factors of the questionnaire that are related to location.” into paragraphs would be better.

Response 6: Thank you very much for your comment. The text has been reread comparing the main results with previous research, even though the research is quite new in this field, thus making comparison difficult. An attempt has been made to combine certain phrases to facilitate comprehension.

Reviewer 2 Report

I read your thesis with interest.
I believe that the field of this study will be helpful in improving school education in the future. I wrote some comments below. I hope it will help you revise your manuscript.

Introduction section
- Please describe the reasons for choosing the last cycle of primary school.

Discussion section
- Please describe the limitations of the study.

Conclusions section
-Please describe the strengths of this study.

Author Response

Thank you very much for your contributions that will make this manuscript better.

Introduction section
Please describe the reasons for choosing the last cycle of primary school.

Response: The reasons that led us to choose this stage for the end of the study were described in the introduction.

Discussion section
- Please describe the limitations of the study.

Response: The limitations of the study and the line to be followed in future research have been described.

Conclusions section
-Please describe the strengths of this study.

Response: Thank you. Strong bridges have been described in this study.

Reviewer 3 Report

In this research, we could understand and investigation and analysis of the motivation of students of the last cycle of primary school in the subject of Physical Education. 

The main question in regards to the methods, was there was any permission from the parents to perform the study?  There is no ethical approval mentioned. I believe that such a study, which is examining children between 9-13 without parental permission, cannot be performed even it is informed by the school or other authority.

Please explain the reason here. 

There are some very disturbing mistakes in the abstract, such as:

Line 19- Google with big capita

L20 - Rho-test pls revise

..

There are some language and form correction needs, please use the language editing service.

In the conclusion please detail more what is the message of this manuscript, that is a novum in the scientific field, or why is this manuscript an important piece of material for the readers.

Author Response

Thank you very much for all the contributions. We are sure that you have made this manuscript better. 

Comment 1: In this research, we could understand and investigation and analysis of the motivation of students of the last cycle of primary school in the subject of Physical Education.

The main question in regards to the methods, was there was any permission from the parents to perform the study?  There is no ethical approval mentioned. I believe that such a study, which is examining children between 9-13 without parental permission, cannot be performed even it is informed by the school or other authority.

Please explain the reason here.

Response 1: Answer 1: Thank you very much for your comment and you are absolutely right. The informed consent  described in the procedure and that was attached was addressed to the parents, who would be the ones who, in compliance with article 13.1 of the LOPD regulation, would give their consent to participate in the study. We attach this document.

Comment 2: There are some very disturbing mistakes in the abstract, such as:

Line 19- Google with big capita

Response 2: Thank you. Changed the spelling of Google Forms to proper capitalization.

L20 - Rho-test pls revise

Response: Thank you. Rho has been removed, leaving only the name of the test (Spearman).

There are some language and form correction needs, please use the language editing service.

In the conclusion please detail more what is the message of this manuscript, that is a novum in the scientific field, or why is this manuscript an important piece of material for the readers.

Response 3: The entire document has been revised to make the requested linguistic and formal corrections and the conclusion has detailed the message of the research, pointing out the strong points of this research.

Round 2

Reviewer 1 Report

Good revision!

Reviewer 3 Report

Thank you for the answers, I accept them!